# Intake of Radionuclides in the Trees of Fukushima Forests 1. Field Study

**Chisato Yasukawa [1], Shoko Aoki [1], Miki Nonaka [1], Masateru Itakura [1], Masaharu Tsubokura [2], Kei'ichi Baba [3], Hiroya Ohbayashi [4], Izumi Sugawara [4], Tomoko Seyama [4], Iwao Uehara [4], Rumi Kaida [1], Teruaki Taji [1], Yoichi Sakata [1] and Takahisa Hayashi [1,*]**

[1]  Department of Bioscience, Tokyo University of Agriculture, Tokyo 156-8502, Japan
[2]  Department of Radiation Protection, Minamisoma Municipal General Hospital, Minamisoma, Fukushima 960-1296, Japan
[3]  Research Institute for Sustainable Humanosphere, Kyoto University, Kyoto 611-0011, Japan
[4]  Department of Forest Science, Tokyo University of Agriculture, Tokyo 156-8502, Japan
*   Correspondence: takaxg@nifty.com

**Abstract:** The earthquake and tsunami on 11 March 2011 led to a meltdown followed by a hydrogen explosion at the Fukushima–Daiichi nuclear power plant in Japan, causing the dispersal of abundant radionuclides into the atmosphere and ocean. The radionuclides were deposited onto trees and local residences in aerosol or gaseous forms that were partly absorbed by rain or melting snow. Here, we show that the radionuclides attached to the surfaces of trees, in which some radiocesium was incorporated into the xylem through ray cells and through symplastic pathways. The level of incorporated radiocesium varied based on tree species and age because of the ability of radiocesium to attach to the surface of the outer bark. After four years, the radiocesium level in the forest has been decreasing as it is washed out with rainwater into the sea and as it decays over time due to its half-life, but it can also be continuously recycled through leaf tissue, litter, mulch, and soil. As a result, the level of radiocesium was relatively increased in the heartwood and roots of trees at four years after the event. In private forest fields, most trees were left as afforested trees without being used for timber, although some trees were cut down. We discuss an interdisciplinary field study on the immediate effects of high radiation levels upon afforested trees in private forest fields.

**Keywords:** Fukushima; radiocesium dispersal; radiocesium infiltration; afforested trees; private forests; farmers' relatedness

## 1. Introduction

The earthquake and tsunami on March 11, 2011, led to a meltdown followed by a hydrogen explosion at the coastal Fukushima–Daiichi nuclear power plant in Japan, causing the dispersal of high levels of radionuclides into the atmosphere and ocean, resulting in a large amount of environmental damage [1]. Among the main volatile fission radionuclides, such as Xe-133, I-131, Te-132, Cs-137, and Cs-134, the most serious damaging fallout to date has involved radiocesium and radioiodine [2]. Cs-137 has a long half-life of 30.17 years, despite Cs-134 having a half-life of only 2 years and radioiodine (mainly $^{131}$I) having a short half-life of 8.04 days. The deposition pattern of radiocesium was not identical to that of radioiodine [3,4], likely because radiocesium was mainly dispersed in an aerosol form rather than in the gaseous form taken by radioiodine in the Fukushima fallout. In the Chernobyl fallout, the aerosol dispersed has been found as particles between 0.1 μm and 1.5 μm in size [5]. In Fukushima, the radiation levels were 150 times levels for radiocesium and three times for radioiodine, which are larger than the levels after the atomic bombing of Hiroshima, but one-fifth and one-tenth,

respectively, less than those of Chernobyl [4,6,7]. The radionuclides were dispersed primarily into the northwest region of the power plant and deposited in significant concentrations over approximately 200,000 hectares of forested rural land in Fukushima, as shown in Figure 1. Half of the land covered private fields, although most forests surrounding Chernobyl were not private fields at the time of the 1986 disaster but were instead under the control of the Soviet Union.

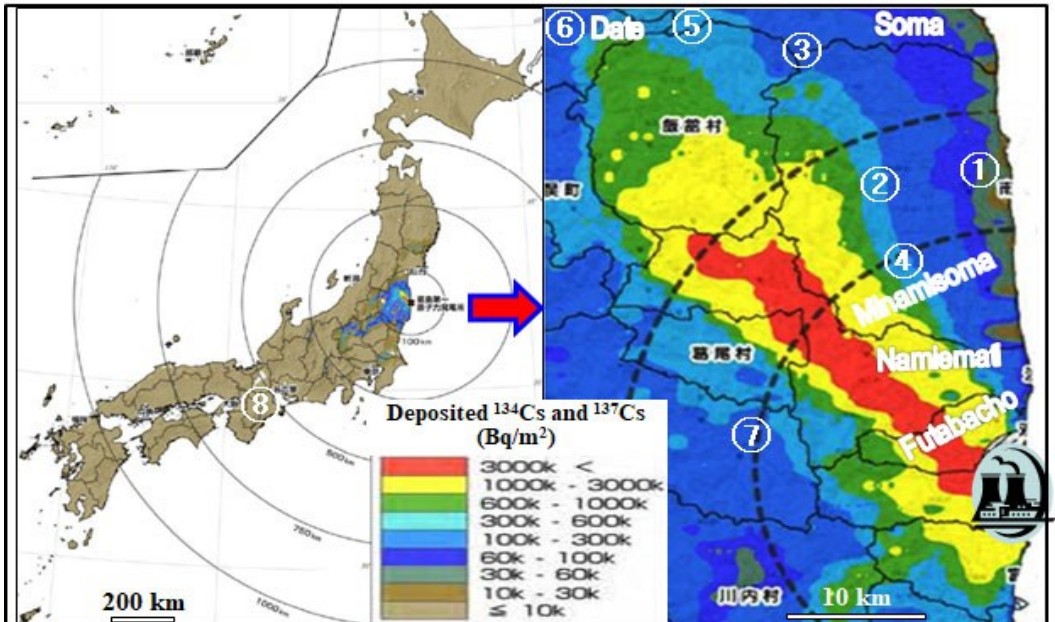

**Figure 1.** Exposure area of eastern Japan showing the rural land where radiocesium ($^{134}$Cs and $^{137}$Cs) was deposited [3,4]. Area numbers: 1, the hospital of Minamisoma; 2 to 7, the places from which tree samples were obtained and where tree seedlings were planted; 8, the place from which control trees were obtained.

The radionuclides that were dispersed were directly deposited onto forest trees, houses, and cars, and also on humans. There was no rain on March 11 and none until the evening of March 15, and high doses of radiation affected the region. For example, on March 14, a Geiger–Muller (GM) counter showed that patients tested at the hospital of Minamisoma, as shown in Figure 1 (area number 1), 23 km from the power plant, exhibited levels of over 13,000 counts per min (cpm), which far exceeded the safe exposure limit, with some patients' levels exceeding 100,000 cpm, as shown in Table 1. Nothing is known about the types of radionuclides that were deposited on the people. Nevertheless, in humans, radiocesium spreads to muscle tissues in the same way as potassium [8], and radioiodine binds to thyroid tissues [9,10], but human exposure is short-term because of the short half-life of these radionuclides in the body [11,12]. Nevertheless, almost nothing is known about radionuclide infiltration into forest trees. The first question is how the radionuclides were incorporated and moved into tree bodies, and the second question is what happened to forest farmers as a result of the dispersal of radionuclides in the private fields. This paper discusses a field study, using both natural and social sciences, in the private forests that were contaminated upon radiocesium being incorporated into trees after the meltdown followed by the hydrogen explosion at the Fukushima nuclear power plant.

**Table 1.** Exposure levels of people tested at the hospital in Minamisoma (Figure 1, area number 1) on 14 March 2011.

| Location of Home | Location of Workplace | Radioactivity |
|---|---|---|
| | | cpm |
| Minamisoma | at home | 13,000 |
| Futabamachi | at home | 22,000 |
| Namiemati | at home | 30,000 |
| Namiemati | at home | 98,000 |
| Minamisoma | at home | greater than 100,000 |
| Minamisoma | at home | greater than 100,000 |
| Minamisoma | Minamisoma | greater than 100,000 |
| Minamisoma | at home | greater than 100,000 |
| Minamisoma | Futabacho | greater than 100,000 |

The locations of homes and workplaces refer to the locations where the people lived and worked, respectively, in the rural areas of Fukushima from 12 to 14 March 2011.

## 2. Materials and Methods

### 2.1. Sampling Wood

All trees were obtained from private plantation forests in Fukushima. Trees that were dominant were cut down at each site shown in Figure 1. Overall, ten species that were major species in each field were sampled from six different forest areas. The mulberry (*Morus alba,* L.) in area number 2, as shown in Table 2, was obtained at 37°38′18″ N/140°54′27″ E. The Japanese cedars (*Cryptomeria japonica* D. Don.) in area numbers 2 (three trees), 4, and 5 were obtained at 37°38′12″ N/140°54′19″ E, 37°38′29″ N/140°54′41″ E, 37°45′50″ N/140°50′26″ E, 37°31′50″ N/140°56′34″ E, and 37°56′16″ N/140°43′33″ E, respectively. The cypress (*Chamaecyparis obtuse* Endl.) in area numbers 2 (two trees), 3, and 4 were obtained at 37°38′12″ N/140°54′19″ E, 37°38′29″ N/140°54′41″ E, 37°46′2″ N/140°52′57″ E, and 37°31′50″ N/140°56′32″ E, respectively. The chestnut (*Castanea sativa* Mill.) in area number 2 was obtained at 37°38′8″ N/140°54′21″ E. The stone oak (*Lithocarpus edulis* (Mak.)) in area number 2 was obtained at 37°38′15″ N/140°54′17″ E. The cherry tree (*Prunus serrulata* L. var. *spontanea*) in area number 2 was obtained at 37°38′38″ N/140°54′55″ E. The persimmon (*Diospyros lotus* L.) in area number 6 was obtained at 37°50′18″ N/140°33′18″ E. The oak (*Quercus* L.) in area number 7 was obtained at 37°24′51″ N/140°47′43″ E. The air levels were 1.50 to 3.00 μSv/h in the forests, except for 0.20 μSv/h around the persimmon in area number 6, 5.50 μSv/h around the cherry tree in area number 2, and 12.0 μSv/h around the Japanese cedar in area number 4.

### 2.2. Planting

One hundred seedlings were planted (20 seedlings from each of the following five species): Japanese cedar (*Cryptomeria japonica*), sawtooth oak (*Quercus acutissima* Carruth.), chinaberry (*Melia azedarach* L.), weeping willow (*Salix babylonica* L.), and white poplar (*Populus alba* L.). The trees were planted with $2 \times 2$ m spacing in May 2012 in a 10,000 m$^2$ field, receiving a 2.00 to 2.05 μSv/h dose rate in the forest (at 37°38′16″ N/140°54′13″ E, Figure 1: area number 2). Some of the trees in this forest had been cut down in November 2012 and November 2013. The total radiocesium ($^{134}$Cs and $^{137}$Cs) in this field was 1.21 MBq/m$^2$, most of which was present in litter and mulch (80%) and soil (20%). It should be noted that all the seedlings were obtained from Hanahiroba (Kuwana, Mie, Japan; Figure 1: area number 8).

### 2.3. Determination of Radiocesium in Trees

Trunk sections were cut in half transversely, and 1cm thick sections were subjected to autoradiography with an image analyzer (Fujifilm, Tokyo, Japan). Image plates were attached to the sample sections for 3 weeks. Under these conditions, no signals were observed from the stored trunks that were sampled before the accident.

The trunk sections (about 3 cm thick, and with above-ground heights of about 1.5 to 2.0 m) were cut transversely and their bark and xylem were analyzed. Each cross-section was longitudinally cut



into two radial directions at right angles to obtain four radial sections (2 cm width) from pith to bark, in which annual growth rings were clearly visible as parallel lines. The four radial sections were mixed and subjected to the determination of radioactivity, and their average levels were shown. Each fraction was powdered, and the dry weight of the tissue was measured after 12 h at 105 °C in a glass vial. The radiation in each fraction was determined using either an Aloka AccuFles γ7001 scintillation counter (Aloka, Tokyo, Japan) or a Beckman LS Gamma Counter (Long Island Scientific, East Setauket, USA), and low radioactivity was verified using Canberra germanium detectors (Canberra, Meriden, USA). The potassium content was determined using an Atomic AA-7000 Absorption Spectrophotometer (Shimazu, Kyoto, Japan) at a wavelength of 766.5 nm after digestion with $HNO_3$ and $H_2O_2$.

**Table 2.** Fall out radiocesium ($^{134}$Cs and $^{137}$Cs) specific activities in the trees of Fukushima forests.

| Sampling Time Area Number * Tree Species (years) | Bark | | Xylem Year | | | | | |
|---|---|---|---|---|---|---|---|---|
| | Outer | Inner | 2012 | 2011 | 2010 | 2009 | 2008 | 2007 |
| | | | kBq/Kg dry Weight | | | | | |
| September 2011 | | | | | | | | |
| 2 Mulberry (9) | 113.95 | 7.10 | | 3.22 | 2.00 | 1.14 | 0.90 | 0.63 |
| 2 Japanese cedar (18) | 42.26 | 14.92 | | 5.43 | 3.86 | 2.64 | 1.83 | 1.37 |
| 2 Cypress (24) | 51.28 | 8.43 | | 2.92 | 1.57 | 1.54 | 1.42 | 0.94 |
| November 2011 | | | | | | | | |
| 2 Chestnut (10) | 75.63 | 4.70 | | 1.59 | 1.41 | 1.45 | 1.25 | 0.53 |
| 2 Japanese cedar (12) | 13.70 | 1.50 | | 1.63 | 1.43 | 1.48 | 1.38 | 1.30 |
| 2 Japanese cedar (21) | 12.92 | 1.22 | | 0.53 | 0.31 | 0.21 | 0.22 | 0.22 |
| 2 Stone oak (8) | 14.91 | 6.55 | | 3.92 | 2.70 | 1.82 | 1.67 | 1.52 |
| 3 Cypress (17) | 3.84 | 1.62 | | 0.38 | 0.08 | 0.14 | 0.14 | 0.09 |
| March 2012 | | | | | | | | |
| 2 Cypress (22) | 10.13 | 5.64 | | 0.77 | 0.45 | 0.38 | 0.44 | 0.38 |
| 2 Cherry tree (30) | 153.06 | 3.05 | | 1.65 | 0.66 | 0.59 | 0.59 | 0.50 |
| 4 Japanese cedar (20) | 43.94 | 3.88 | | 0.47 | 0.29 | 0.28 | 0.30 | 0.36 |
| 4 Cypress (19) | 9.36 | 4.50 | | 0.89 | 0.27 | 0.47 | 0.46 | 0.56 |
| November 2012 | | | | | | | | |
| 5 Japanese cedar (19) | 72.07 | 10.08 | 3.84 | 0.50 | 0.63 | 0.44 | 0.49 | 0.56 |
| 6 Persimmon (18) | 0.75 | 1.33 | 0.70 | 0.10 | ND | ND | ND | ND |
| 7 Oak (21) | 6.63 | 0.19 | 0.10 | ND | ND | ND | ND | ND |

* Area numbers are shown in Figure 1. All the vascular cambium zones were removed. Individual values represent the means of three independent samples for each year ring with individual values varying from the mean by <5.1%. ND, not detected.

### 2.4. Microscopy for Radiocesium Autoradiography

Japanese cedar (*Cryptomeria japonica*) was obtained from the forest (at 37°38′16″ N/140°54′13″ E, Figure 1: area number 2) on September 2012. High levels of radionuclides in the bark and xylem were a result of radiocesium $^{134}$Cs and $^{137}$Cs, in which the radioactivity in the cut section was determined to be about 71 kBq/kg using the Canberra germanium detectors.

The 5 mm sections (0.5 × 0.5 × 0.5 cm) were dipped twice in *n*-butanol for 3 min each and embedded in paraffin at 70 °C, according to the method described by Edelmann [8]. Paraffin-embedded stems were transversely sliced using a micro slicer, and the cut sections were then placed onto glass slides and washed with xylene to remove the paraffin. The sections mounted on glass slides were covered with Ilford K5D emulsion film (Ilford Photo, Cheshire, UK) and stored at 4 °C for 3 months. They were developed using Kodak D-19 (Kodak, Rochester, USA) for approximately 5 min, rinsed with water, fixed with Kodak fixer at 20 °C, washed in running water (15 min), and sealed with 50% glycerol. Slides were examined under bright-field illumination and photographed. Control Japanese cedar stems were obtained from Hanahiroba (Kuwana, Mie, Japan; Figure 1: area number 8). The glass slides with the bark and xylem of Japanese cedar were stored at 4°C for 3 months.

### 2.5. Forest Analysis

Sampling was performed over several days in April 2012 and November 2015 in the forest (at 37°38′11″ N/140°54′18″ E, Figure 1: area number 2) located about 30 km northwest of the Fukushima–Daiichi nuclear power plant. The air dose was 2.00 to 2.50 μSv/h in April 2012 and

0.80 μSv/h in November 2015. There were about 1750 trees in the private forest, including 1500 Japanese cedar (15 to 25 m tall) (*Cryptomeria japonica*), 150 cypress (10 to 15 m tall) (*Chamaecyparis obtusa*), and 100 momi fir (*Abies firman* Sieb. & Zucc.). The height and diameter of the trees at 150 cm above ground and their volumes were surveyed and calculated from the stem volume tables of each species [13]. The forest field was divided into $2 \times 2$ m plots using white ropes on the surface field, in which samples of soil, litter (leaves and branches on the soil surface), grass plants, insects, and animals were collected and precisely measured for each 2 m square section in the 10,000 $m^2$ field. Soil was taken at 1 cm layers to a 10 cm depth to obtain 10 layer fractions.

Four representative trees of the dominant species, including roots, were completely dug up. They were then divided into their structural components, such as young needles of the current year, old needles, foliage, large and small branches, generative organs, outer bark and inner bark, wood, and roots, and these samples were used to determine the average distribution of radiocesium in the trees in each plot. For each structural component, radioactivity was determined in at least 12 samples. Additionally, the upper, middle, and lower parts of stems were transversely cut into about 3 cm thicknesses. Each cross-section was longitudinally cut to produce about four sections (about 2 cm width), in which annual growth rings were clearly visible as parallel lines. Radioactivity was determined in the four sections, and their average levels were shown. The radiation in each fraction was determined using an Aloka AccuFles γ7001 scintillation counter (Aloka), and further determined using Canberra germanium detectors (Canberra).

Wood samples were also obtained in December 2012 in the forest (at 37°38′17″ N/140°54′19″ E, Figure 1: area number 2) that was located about 30 km northwest of the Fukushima–Daiichi nuclear power plant. At that point, the air dose was 1.00 to 1.50 μSv/h. Most of the trunk samples used were from 1.5 to 2.0 m above ground. Each trunk was cut transversely and fractionated into outer bark, inner bark, sapwood, and heartwood. The radioactivity of each tissue was determined using Canberra germanium detectors.

## 3. Results

### 3.1. Effects of Radiocesium Fallout on the Forest Trees

Our field sites (Date, Soma, Minamisoma, and Tamura) are principally located in the central-to-far eastern parts of the region, which coincides with the highest levels of radiocesium fallout (ambient dosage, 0.2 to 12.0 μSv per h), as shown in Figure 1. The radiocesium ($^{134}$Cs and $^{137}$Cs) was found at high levels on the outer bark of mulberry (19 years) and cherry (30 years) trees, as shown in Table 2, located on mountainsides facing the radiation epicenter, which were relatively high in elevation (more than 400 m above sea level). These data indicate that accumulation was continuous in the trees in the region between the power plant and the mountains. The radiation was absorbed by the outer and inner bark and the xylem. It should be noted that no deciduous trees had leaves in March 2011.

As shown in Table 2, a high level of radiocesium migration into the sapwood was observed in Japanese cedar (18 years), cypress (24 years), and stone oak (8 years). For Japanese cedar (21 years) and cypress (19 years), the heartwood showed high levels of radiation (1.10 and 1.30 kBq/kg, respectively) despite the relatively low levels in the bark and sapwood. However, persimmon (18 years) and oak (21 years) trees retained most of the radiation in the bark.

In the 2 year old sucker sprout seedlings from tree stumps in the forest after the accident, as shown in Figure 1 (area number 2), the radiocesium in the sprout was efficiently translocated to the leaves, branches, and xylem, but only a small amount was located on the bark in November 2012 and June 2013, as shown in Table 3. Five kinds of seedlings were also planted there on March 2012. After one and a half years, the amounts of radiocesium were lower in the bark of seedlings compared to the soil and in the inner bark of trees that were standing before the accident.

**Table 3.** Incorporation of radiocesium ($^{134}$Cs and $^{137}$Cs) into trees either developed from the stumps of trees or planted in the forest after the accident (Figure 1: area number 2). The total radiocesium ($^{134}$Cs and $^{137}$Cs) in this field was 1.21 MBq/m$^2$, most of which was present in litter and mulch (80%) and soil (20%).

| Tree Species | Bark | Xylem | Leaf and Branch |
|---|---|---|---|
| Bq/Kg dry Weight | | | |
| Sprouted shoots from the stumps of trees (Sampling time in November 2012) | | | |
| Cypress | 100 ± 2.0 | 630 ± 6.6 | 2800 ± 80.9 |
| Sawtooth oak | 110 ± 2.1 | 540 ± 10.1 | 2250 ± 56.5 |
| Planted seedlings planted on May 2012 (Sampling time in November 2012) | | | |
| Chinaberry | 68 ± 1.1 | 430 ± 4.1 | 1310 ± 7.0 |
| Sawtooth oak | 50 ± 1.0 | 200 ± 2.0 | 530 ± 4.5 |
| White poplar | 41 ± 05 | 430 ± 2.2 | 1030 ± 5.8 |
| Japanese cedar | 57 ± 0.5 | 220 ± 3.5 | 670 ± 4.9 |
| Weeping willow | 10 ± 0.1 | 140 ± 2.3 | 670 ± 5.6 |
| (Sampling time in June 2013) | | | |
| Chinaberry | 81 ± 1.1 | 720 ± 4.1 | 1560 ± 7.0 |
| Sawtooth oak | 76 ± 1.0 | 610 ± 2.0 | 1350 ± 4.5 |
| White poplar | 88 ± 0.5 | 630 ± 2.2 | 1110 ± 5.8 |

Each data point represents the means of three independent samples of trees for each line, with individual values varying from the mean by less than 2.0%.

### 3.2. Distribution of Radiocesium in Bark and Xylem

The typical radiocesium distribution in xylem is illustrated in the autoradiographs in Figure 2, which includes images for chestnut (10 years) and Japanese cedar (12 years) that show the migration of the radionuclide into the sapwood, and the other image for Japanese cedar (21 years) shows its migration to the heartwood. The high levels of radiocesium might be attributed to retention by the outer bark. The alkali metal ion appeared to enter gradually via the lenticels and loose outer bark from the surface of the bark through the inner bark and into the xylem. As shown in Figure 2, the localization of radiocesium in the sapwood could be related to that found in the inner bark of the Japanese cedar (12 years).

Although two Japanese cedar trees from the same forest area were tested at the same time, one showed radiocesium migration into the sapwood and the other into the heartwood, as shown in Figure 2. The distribution of radionuclide in the tissues seems to be slowed down by the layer of intermediate wood between the sapwood and heartwood. The alkali metal ions appear to move to the heartwood if the level of potassium in a tree is sufficient in the growing region. The level (1.71 mg per kg wood) in the outer ring of the 21 year old sapwood was much higher than that of the 12 year old sapwood (1.07 mg per kg wood) and that of the chestnut (1.11 mg per kg wood). Excess potassium ions seem to be forced to move with radiocesium ions as excretory metal ions into the heartwood, coloring the heartwood much darker than its typical red coloring.

In the transverse stem sections of the Japanese cedar (12 years), as shown in Figure 2A, the silver grains indicating $^{134}$Cs and $^{137}$Cs were localized in the cork cambium between the outer and inner bark layers, as shown in Figure 2B. In the xylem, the grains were also observed in ray cells and in the narrow tips of the tracheids, where pit membranes are located between the narrow tips. The alkali metal ions could be transported longitudinally by the apoplastic pathway, such as vessels and tracheids in tree plants, although grass plants use the phloem via the symplastic pathway [14]. The levels of radiocesium were higher in the xylem than in the bark in young tree seedlings planted in the forest after the accident, as shown in Table 3.

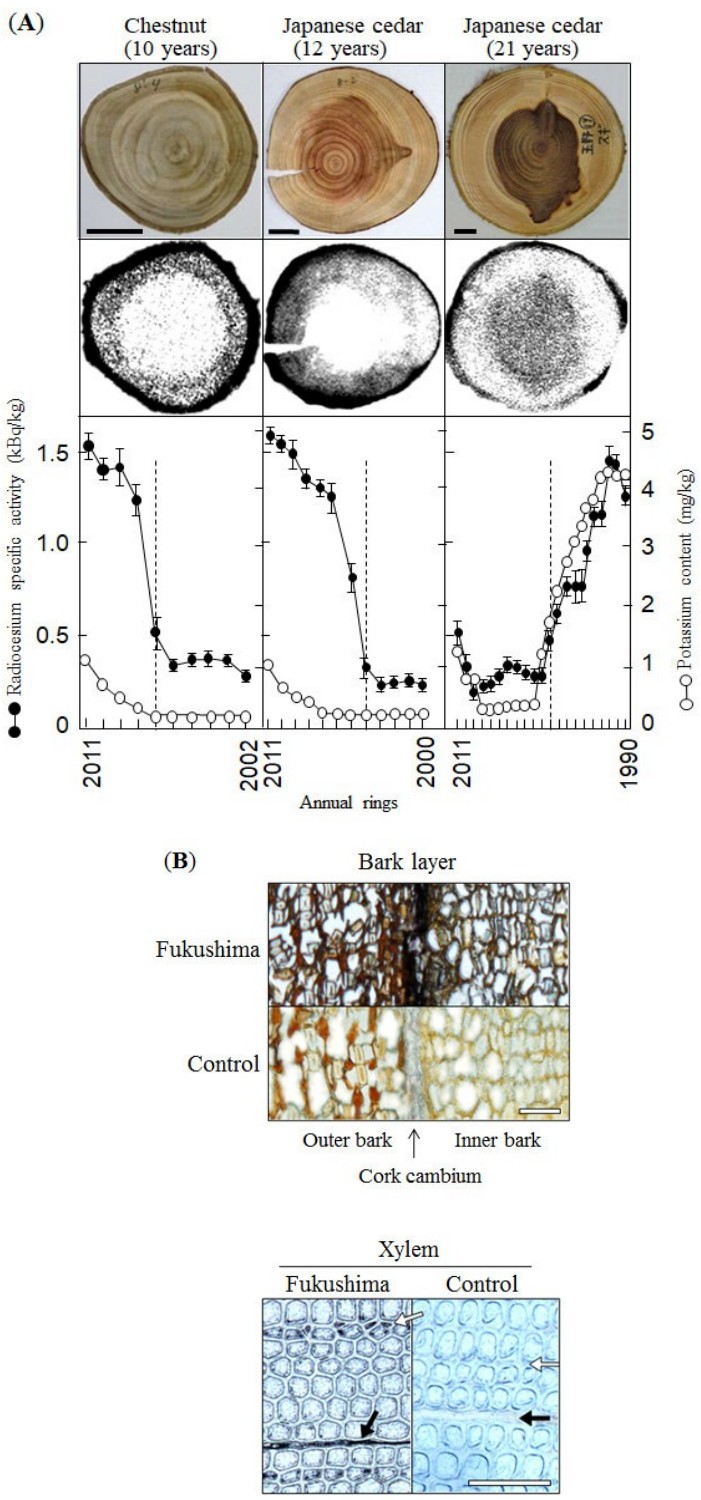

**Figure 2.** Transverse sections of stems with autoradiographs and levels of radiocesium and potassium. (**A**) Chestnut stem (Figure 1: area number 2), Japanese cedar stem (Figure 1: area number 2), and Japanese cedar stem (Figure 1: area number 2). Radiocesium ($^{134}$Cs and $^{137}$Cs) and potassium levels were determined in 2011 wood from the left side of each ring to the pith, from which the vascular cambium zone was removed. Dotted lines show the intermediate wood between the sapwood and heartwood. Individual values represent the mean ± SD of three independent measurements. Bars = 2 cm. (**B**) Transverse sections of the bark layer and xylem of a Japanese cedar stem from the forest, one year after the nuclear accident (upper) and a control section (lower). In xylem, the black arrow shows ray cells, and the white arrow shows the narrow tips of the tracheids. Bars = 100 μm.

As shown in Table 4, the heartwood in approximately 17% of trees in the forest exceeded the permissible level of 1.00 kBq/kg, which was set by the Ukrainian government after the Chernobyl accident for forestry products such as timber [15]. It is possible that radiocesium gradually accumulates in the heartwood of trees and then continues to affect forest products.

**Table 4.** Radiocesium ($^{134}$Cs and $^{137}$Cs) specific activities in trees in the forest (Figure 1: area number 2) on December 2012.

| Tree Species (Years) | Ambient Dose | Bark | | Xylem | |
|---|---|---|---|---|---|
| | | Outer | Inner | Sapwood | Heartwood |
| | μSv/h | kBq/kg Dry Weight | | | |
| Japanese cedar (13) | 2.45 | 8.14 ± 0.32 | 3.34 ± 0.14 | 2.13 ± 0.13 * | 2.60 ± 0.04 * |
| Japanese cedar (13) | 2.00 | 8.90 ± 0.41 | 5.66 ± 0.19 | 0.57 ± 0.02 * | 1.25 ± 0.03 * |
| Japanese cedar (18) | 1.88 | 8.37 ± 0.32 | 7.29 ± 0.22 | 0.87 ± 0.04 * | 1.10 ± 0.03 * |
| Japanese cedar (20) | 1.90 | 14.22 ± 1.00 | 4.55 ± 0.12 | 0.58 ± 0.03 * | 0.99 ± 0.03 * |
| Japanese cedar (29) | 2.00 | 11.36 ± 1.04 | 4.52 ± 0.11 | 0.63 ± 0.05 | 0.86 ± 0.03 |
| Japanese cedar (16) | 2.45 | 12.81 ± 0.92 | 2.29 ± 0.08 | 0.36 ± 0.02 | 0.69 ± 0.03 |
| Red pine (32) | 1.64 | 6.01 ± 0.22 | 7.87 ± 0.33 | 1.48 ± 0.23 | 0.56 ± 0.02 |
| Pine (41) | 2.36 | 14.32 ± 1.11 | 3.22 ± 0.12 | 0.65 ± 0.02 * | 0.51 ± 0.01 |
| Red pine (46) | 1.62 | 14.11 ± 1.22 | 5.72 ± 0.25 | 1.09 ± 0.32 | 0.49 ± 0.01 |
| Cypress (15) | 2.54 | 4.84 ± 0.32 | 3.23 ± 0.17 | 0.53 ± 0.02 | 0.48 ± 0.02 |
| Japanese cedar (30) | 2.01 | 6.05 ± 0.52 | 4.00 ± 0.18 | 0.51 ± 0.01 | 0.42 ± 0.02 |
| Japanese cedar (14) | 2.44 | 11.34 ± 0.94 | 4.42 ± 0.17 | 2.66 ± 0.25 | 0.41 ± 0.01 * |
| Red pine (27) | 1.60 | 7.80 ± 0.33 | 6.86 ± 0.22 | 1.01 ± 0.16 | 0.40 ± 0.01 |
| Japanese cedar (35) | 2.00 | 8.61 ± 0.40 | 4.87 ± 0.22 | 0.35 ± 0.02 | 0.30 ± 0.01 |
| Cypress (14) | 2.10 | 8.65 ± 0.41 | 5.57 ± 0.27 | 1.22 ± 0.01 | 0.30 ± 0.01 |
| Japanese cedar (43) | 2.02 | 9.54 ± 0.45 | 3.55 ± 0.22 | 0.28 ± 0.01 | 0.30 ± 0.01 |
| Cypress (22) | 2.02 | 10.07 ± 0.95 | 6.54 ± 0.30 | 1.38 ± 0.07 | 0.21 ± 0.01 |
| Cypress (16) | 2.23 | 8.58 ± 0.33 | 7.19 ± 0.32 | 0.37 ± 0.01 | 0.05 ± 0.01 |

All the vascular cambium zones were removed. *$p < 0.05$ based on Student's t-test and data are means ± SD.

### 3.3. Radionuclides in The Forests

In April 2012, one year after the accident, a very low level of radioiodine (2 to 5 Bq as mostly $^{131}$I) was detected, but high levels of $^{134}$Cs and $^{137}$Cs were detected at levels similar to those in the forest, as shown in Figure 1 (area number 2). The total radiocesium in this forest was 1.77 MBq/m$^2$, and the distribution was as follows: litter and mulch (72%), trees (14%), and soil (13%). The trees contained radionuclides in the bark (42%), leaves (25%), xylem (13%), branches (13%), and roots (7%), as shown in Figure 3. Some of the radionuclides in the litter replaced those in the leaves because some of the leaves exposed in 2011 were recovered from the litter and mulch fraction. Because most of the radiocesium in trees was present in the bark, the trees' major source of irradiation might have been direct exposure.

In April 2013, total radiocesium in this field had decreased to 0.98 MBq/m$^2$, mainly because of a decrease in $^{134}$Cs; the ratio of $^{134}$Cs to $^{137}$Cs activity was 0.55. The distribution of the radionuclides was relatively increased in xylem, branches, and roots compared to other tissues.

In November 2015, total radiocesium in this field had decreased to 0.36 MBq/m$^2$, mainly because of a marked decrease in $^{134}$Cs to one-fifth and a gradual decrease in $^{137}$Cs to five-sixths of the original amounts. A further decrease in the level was due to dispersion outside of the forest. The $^{134}$Cs/$^{137}$Cs activity ratio was 0.20, and $^{137}$Cs became the main radiocesium isotope. The distribution of the radionuclides was relatively increased in the soil but remained constant in the trees. It was also relatively increased in the xylem and roots compared to the other tissues.

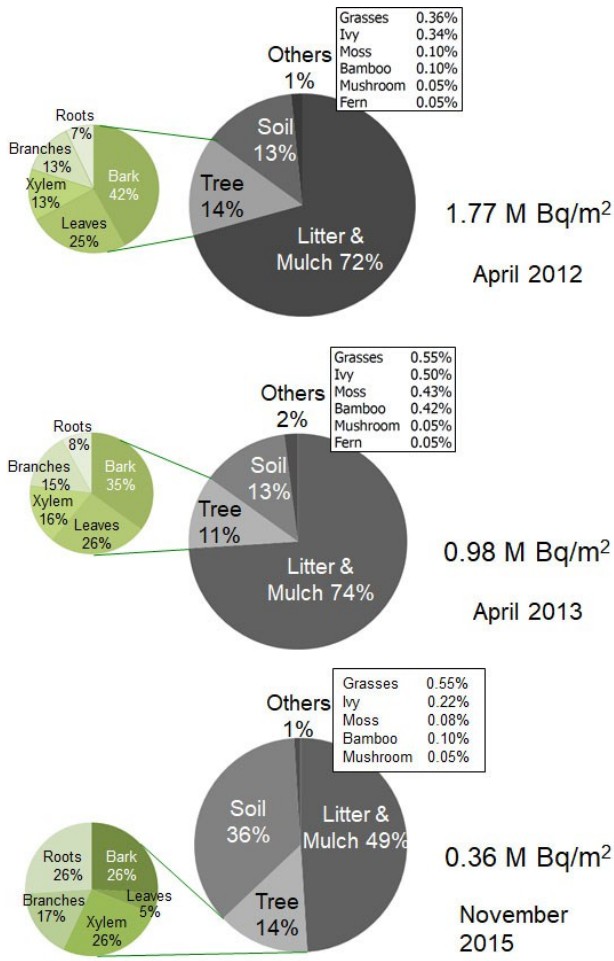

**Figure 3.** Distribution of radiocesium ([134]Cs and [137]Cs) in the forest field (Figure 1: area number 2) from April 2012 to November 2015 after the nuclear plant accident. Each percent content represents the mean value of three independent fractions, with individual percentage values varying from the mean by less than 1.1%.

## 4. Discussion

Dispersed radiocesium attached to the surfaces of tree stems, the levels of which were a result of their ability to attach to the outer bark. Radiocesium then migrateed on the surface of the outer bark, and, in this manner, the radionuclides might enter the lenticels and move from the loose outer bark into the inner bark. The radiocesium was gradually translocated to the xylem, where radiocesium levels were higher in either the sapwood or the heartwood, depending on the potassium concentration. The radiocesium moved transversely from the bark to the sapwood and to the heartwood via ray cells and also longitudinally via tracheids, and then to the new branches, their xylem, and leaves. However, radiocesium levels were relatively lower in the bark of newly formed branches and the seedlings planted in the forest after the accident, as shown in Table 3. Nevertheless, high levels of radiocesium have been recycling in Fukushima forests [16,17] and in some other areas [18] in Japan, and also at lesser amounts around the world [19] because of the global spread of the Fukushima aerosols [20]. The forests in rural areas have become enormous unnatural repositories of [137]Cs, which has a long half-life of 30.17 years, despite [134]Cs having a half-life of 2 years. [137]Cs in the leaf litter and mulch can become a soil component, and some of the radiocesium can then be re-incorporated into the trees, as observed after the Chernobyl disaster [21,22]. Although most of the radionuclides in forests are recycled among living plant tissues, litter, mulch, and soil, some continue to accumulate in the heartwood of trees.

A permissible level of radionuclides in wood has not been set in Japan since the Fukushima nuclear power plant accident, but a safety declaration was performed by the forestry agency in Japan on August 2012 [23]. However, the Ukrainian government set less than 1000 Bq/kg as a permissible level for wood after the Chernobyl accident because they believed that highly contaminated forest products should not be distributed to other places. A biomass energy company in Sweden has carefully checked wood and bark transported from various Swedish forests because it is difficult for their plant system to capture radiocesium flyash from more than 1000 Bq/kg biomass sources even after 30 years [24].

We asked the Japanese Forestry Agency to set the permissible level for wood on October 2012 because the level for food had been set at less than 100 Bq/kg. We also showed that high levels of radiocesium had been deposited in some wood in Fukushima. The forestry agency replied that it was not necessary to set the permissible level for wood in Japan at that time, but that it would be possible in the future. The following month, in November 2012, we presented our results showing the intake of radiocesium in the trees from the Fukushima forests at a symposium in which many politicians and members of Parliament were present [25]. One of the members of the House of Representatives Democratic Party of Japan explained that the reason why the government had not set the level might be because of disagreement from the "mokuzai-mura", which means "wood village," similar to the "nuclear power village" in Japan, as reported by Onishi and Belson [26]. What does the village ("mura") refer to? A nuclear power village refers to a special social group formed by specific stakeholders from industry, government, and academia, who are connected by the interests surrounding nuclear power generation. It is said to be a profit community centered on industrial enterprises in a village-like manner. It is likely that Japanese wood industries were afraid of a harmful rumor about contaminated wood in Fukushima and also about all wood in Japan.

A rule was established that prohibited the transportation of timber with more than 15,000 Bq/kg on any public road in Fukushima. Additionally, the sale of wood with more than 1000 cpm on its surface, as measured by a survey meter, was prohibited. Therefore, the forest cooperatives could not harvest the highly contaminated trees because of these rules that had been implemented, and the wood was checked using a survey meter to see if it contained high levels of radionuclides. However, forest farmers were allowed to harvest trees, which could then be transported and sold by private companies. Given the transportation rule, it is unknown whether these companies were able to take advantage of this loophole and sell the timber to other places in Japan. In fact, some of their fields have become wastelands after harvest, where the forest stock has not been replenished through replanting because the farmers understand that any seedlings that they plant could re-incorporate radiocesium and generate contaminated timber in the future.

To date, nobody has burned their private forests to receive compensation from their forest insurance, but wildfires often occurred around the Chernobyl nuclear plant [27,28]. The old and young forest farmers have a relationship with the trees, and the farmers can harvest trees that were planted by their great-grandfathers. This relatedness could also prevent them from burning their forest because the trees were planted by their forebears. Even after cutting down the trees, this relatedness could prevent them from planting tree seedlings in the deforested fields because they would not want their legacy to be contaminated trees for their children and grandchildren. Nevertheless, mothers with children and many young families have moved away from the rural land because of the fear of radiation exposure and the lack of local jobs, although old farmers may stay for the forest, even with high levels of radiation in the air. Additionally, many frail elderly people live alone in their houses or in temporary housing, which is associated with low to high levels of radiation.

## 5. Conclusions

The earthquake and tsunami on 11 March 2011, led to a succession of accidents at the Fukushima-Daiichi nuclear power plant in Japan, causing the dispersal of abundant radionuclides into the atmosphere, falling down on trees and local residences. To date, nobody has set fire to their private forests to get a compensation due to their forest insurance. Because there is a relatedness between the

old and young forest farmers on the trees planted by their great-grandfathers. However, mothers with children and many young families moved out of the rural land due to the fear of radiation exposure and the lack of local jobs, although old farmers would stay and work for the forest even at high levels of air dose. What can be said in the disaster is such that the forest trees of Fukushima could have saved the residents of Fukushima from worse contamination, much larger than the levels of the radionuclides infiltrated into people.

**Author Contributions:** Conceptualization and methodology, T.H.; investigation, C.Y., S.A., M.N., M.I., M.T., K.B., H.O., I.S., T.S., I.U., R.K., T.T., Y.S., and T.H., writing, T.H., funding acquisition, T.T., Y.S., and T.H.

**Funding:** This work was supported by a grant from Tokyo University of Agriculture for the Eastern Japan Reconstruction Support Project after the Fukushima Disaster.

**Acknowledgments:** We thank M. Jacobs for his comments regarding English-language nomenclature, Little for comments on the forest disaster, O. Masson for radiology comments, and Y. Kono for social science comments on the manuscript. We are grateful to Y. Takeyama for allowing us to use the plantation field in the Minamisoma forest, to Y. Hoshi for allowing us to conduct the forest analysis in the Minamisoma forest, to E. Horiuchi and S. Sasaki for collecting samples, and to the foresters in Soma and Minamisoma for help with the fieldwork. We thank H. Umetsu, A. Hasegawa, S. Mori, Y. Kobayashi, M. Takahashi, K. Naganawa, S. Takeshita, Y. Hirosue, and I. Shiraishi for their technical support during the fieldwork.

**Conflicts of Interest:** The authors declare no conflict of interest.

## References and Note

1.　Fukushima Daiichi Nuclear Disaster. Wikipedia 2019. Available online: https://en.wikipedia.org/wiki/Fukushima_Daiichi_nuclear_disaster (accessed on 11 November 2014).

2.　Koo, Y.H.; Yang, Y.S.; Song, K.W. Radioactivity release from the Fukushima accident and its consequences: A review. *Prog. Nucl. Energy* **2014**, *74*, 61–70. [CrossRef]

3.　Torii, T.; Sugita, T.; Okada, C.E.; Reed, M.S.; Blumenthal, D.J. Enhanced analysis methods to derive the spatial distribution of $^{131}$I deposition on the ground by surveys at an early stage after the Fukushima Daiichi nuclear power plant accident. *Health Phys.* **2013**, *105*, 192–200. [CrossRef] [PubMed]

4.　Achim, P.; Monfort, M.; LePetit, G.; Gross, P.; Douysset, G.; Taffary, T.; Blanchard, X.; Moulin, C. Analysis of radionuclide releases from the Fukushima Dai-ichi Nuclear Power Plant accident Part II. *Pure Appl. Geophys.* **2014**, *171*, 645–667. [CrossRef]

5.　Jost, D.T.; Gaggeler, H.W.; Baltensperger, U. Chernobyl fallout insize-fractionated aerosol. *Nature* **1986**, *324*, 22–23. [CrossRef] [PubMed]

6.　Yasunari, T.J.; Stohl, A.; Hayano, R.S.; Burkart, J.F.; Eckhardt, S.; Yasunari, T. Cesium-137 deposition and contamination of Japanese soils due to the Fukushima nuclear accident. *Proc. Natl. Acad. Sci. USA* **2011**, *108*, 19530–19534. [CrossRef] [PubMed]

7.　LePetit, G.; Douysset, G.; Ducros, G.; Gross, P.; Achim, P.; Monfort, M.; Raymond, P.; Pontillon, Y.; Jutier, C.; Blanchard, X.; et al. Analysis of radionuclide releases from the Fukushima Dai-ichi Nuclear Power Plant accident Part I. *Pure Appl. Geophys.* **2014**, *171*, 629–644. [CrossRef]

8.　Edelmann, L. Potassium binding sites in muscle: Electron microscopic visualization of K, Rb, and Cs in freeze-dried preparations and autoradiography at liquid nitrogen temperature using $^{86}$Rb and $^{134}$Cs. *Histochemistry* **1980**, *67*, 233–242. [CrossRef] [PubMed]

9.　Orlando, P.; Gallelli, G.; Perdelli, F.F.; Flora, S.D. Alimentary restrictions and $^{131}$I in human thyroids. *Nature* **1986**, *324*, 23. [CrossRef] [PubMed]

10.　Tokonami, S.; Hosoda, M.; Akiba, S.; Sorimachi, A.; Kashiwakura, I.; Balonov, M. Thyroid doses for evacuees from the Fukushima nuclear accident. *Sci. Rep.* **2012**, *2*, 1–4. [CrossRef]

11.　Tsubokura, M.; Gilmore, S.; Takahashi, K.; Oikawa, T.; Kanagawa, Y. Internal radiation exposure after the Fukushima nuclear power plant disaster. *JAMA* **2012**, *308*, 669–670. [CrossRef] [PubMed]

12.　Hayano, R.S.; Tsubokura, M.; Miyazaki, M.; Satou, H.; Sato, K.; Masaki, S.; Sakuma, Y. Internal radiocesium contamination of adults and children in Fukushima 7 to 20 months after the Fukushima NPP accident as measured by extensive whole-body-counter surveys. *Proc. Jpn. Acad. Ser.* **2013**, *89*, 157–163. [CrossRef] [PubMed]

13. Japan Forest Agency. *Tree Volume Table -East Japan*; Japan Forest Investment Cooperation: Tokyo, Japan, 1980. (In Japanese)

14. Hayashi, H.; Chino, M. Nitrate and other anions in the rice phloem sap. *Plant Cell Physiol.* **1985**, *26*, 325–330.

15. State Hygienic Standard of Ukraine. Hygienic norms of the radionuclides [137]Cs and [90]Sr specific activities in wood and wood products. *Approv. Decree Minist. Healthc. Ukr.* **2005**, *573*. dd. 31.10.

16. Hashimoto, S.; Ugawa, S.; Nanko, K.; Shichi, K. The total amounts of radioactively contaminated materials in forests in Fukushima. *Jpn. Sci. Rep.* **2012**, *416*, srep00416. [CrossRef] [PubMed]

17. Kuroda, K.; Kagawa, A.; Tonosaki, M. Radiocesium concentrations in the bark, sapwood and heartwood of three tree species collected at Fukushima forests half a year after the Fukushima Dai-ichi nuclear accident. *J. Environ. Rad.* **2013**, *122*, 37–42. [CrossRef] [PubMed]

18. Qin, H.B.; Yokoyama, Y.; Fan, Q.H.; Iwatani, H.; Tanaka, K.; Sakaguchi, A.; Kanai, Y.; Zhu, J.M.; Onda, Y.; Takahashi, Y. Investigation of cesium adsorption on soil and sediment samples from Fukushima Prefecture by sequential extraction and EXAFS technique. *Geochem. J.* **2012**, *46*, 297–302. [CrossRef]

19. Perrot, F.; Marquet, C.H.; Pravikoff, M.S.; Bourquin, P.; Chiron, H.; Guernion, P.Y.; Nachab, A. Evidence of [131]I and [134,137]Cs activities in Bordeaux, France due to the Fukushima nuclear accident. *J. Environ. Rad.* **2012**, *114*, 61–65. [CrossRef] [PubMed]

20. Lujaniene, G.; Bycenkiene, S.; Povince, P.P.; Gera, M. Radionuclides from the Fukushima accident in the air over Lithuania: Measurement and modeling approaches. *J. Environ. Rad.* **2012**, *114*, 71–80. [CrossRef]

21. Tikhomirov, F.A.; Shcheglov, A.I.; Sidorov, V.P. Forests and forestry–radiation protection measures with special reference to the Chernobyl accident zone. *Sci. Total Environ.* **1993**, *137*, 289–305. [CrossRef]

22. Thiry, Y.; Colle, C.; Yoschenko, V.; Levchuk, S.; Hees, M.V.; Hurtevent, P.; Kashparov, V. Impact of Scots pine (*Pinus sylvestris* L.) plantings on long term [137]Cs and [90]Sr recycling from a waste burial site in the Chernobyl Red Forest. *J. Environ. Rad.* **2009**, *100*, 1062–1068. [CrossRef]

23. The Forestry Agency, Survey Results of Radiocesium Concentration in Trees. Available online: http://www.rinya.maff.go.jp/j/press/mokusan/120809_1.html (accessed on 20 September 2011). (In Japanese)

24. McCormick, K.; Kaberger, T. Exploring a pioneering bioenergy system: The case of Enkoping in Sweden. *J. Clean. Prod.* **2005**, *13*, 1003–1014. [CrossRef]

25. Hayashi, T. The Role Played by the Forest. Symposium on Exposure/Health Damage by Medical Reform Promotion Council from the Site. 11 November 2012. (In Tokyo).

26. Onishi, N.; Belson, K.; Culture of Complicity Tied to Stricken Nuclear Plant. The New York Times, 26 April 2011. Available online: http://www.nytimes.com/2011/04/27/world/asia/27collusion.html?src=me (accessed on 26 April 2011).

27. Yoschenko, V.I.; Kashparov, V.A.; Protsak, V.P.; Lundin, S.M.; Levchuk, S.E.; Kadygrib, A.M.; Zvarich, S.I.; Khomutinin, Y.V.; Maloshtan, I.M.; Lanshin, V.P.; et al. Resuspension and redistribution of radionuclides during grassland and forest fires in the Chernobyl exclusion zone: Part I. Fire experiments. *J. Environ. Rad.* **2006**, *86*, 143–163. [CrossRef] [PubMed]

28. Yoschenko, V.I.; Kashparov, V.A.; Levchuk, S.E.; Glukhovskiy, A.S.; Khomutinin, Y.V.; Protsak, V.P.; Lundin, S.M.; Tschiersch, J. Resuspension and redistribution of radionuclides during grassland and forest fires in the Chernobyl exclusion zone: Part II. Fire experiments. *J. Environ. Rad.* **2006**, *87*, 260–278. [CrossRef] [PubMed]

