# Peer review of "Intake of Radionuclides in the Trees of Fukushima Forests 1. Field Study"

_forests, doi:10.3390/f10080652_

Round 1

Reviewer 1 Report

A very interesting work and important discussion.

Comment: Line 66. The authors most probably mean a portable scintillation counter. The Geiger-Mueller is not a scintillation detector and, on the other hand, not especially suitable for the counting of gamma radiation.

The caption of the Fig. 2B must be set in the right position.

Fig 3: Is the "Tree" value for April 2013 (11%) correct?

The Author contribution and Funding not complete.

Acknowledgements: "Little for comments on the forest disaster" Initial of Mr/Dr. (?) Little is missing. 

Reviewer 2 Report

The paper deals about the intake of radionuclides in the trees of Fukushima forest. The results are sound and the conclusions are supported by the results. It was overall well written, but I have some comments as below:

2.1. Sampling Wood

Line 86-93 This information must be collected in a table not in the text.

2.3. Determination of radiocesium in trees.

Lines 115- 120 A detailed explanation of the process of preparation of the samples, as well as the efficiency  calibration of the equipment (scintillation counter, gamma counter and germanium detector) is necessary.

2.4. Microscopy for Radiocesium Autoradiography

Line 124 “…and the radioactivity” radioactivity of what?

4. Discussion

Lines 295-332 Unnecessary information in a scientific paper.

Reviewer 3 Report

Look at attached file!

Reviewer 4 Report

 I congratulate the authors to this interesting study, that presents a very nice sampling of forests in the Fukushima district after the fallout in 2011. But I think that the authors do not take the best of this work.

The sampling need to be presenting in a more clear way, by adding tables (sampling date, tree species) in several tables (it could be added in supplementary material.)

Focus on 134/137Cs and forget 131I. It is necessary to provide in the introduction the half live of each isotope and to explain briefly the relative decay. Additional figures need to be added. (see comments below).

Data: in fact, mostly presented in tables and as mean values, it is necessary to provide this mean with the standard deviation. It is a shame that the variability of Cs for the same type of samples is not discussed. It must be added as such measurements are not frequent. It could help further works to estimate the appropriate number replicate.

Reference to forest farmers not very clear in the present form. Some parts were really written for this article or recovered from a report/article on Chernobyl. Must be suppressed or improved.

Add references (see comments)

The present version is more a report than an article, it needs to be improved.

Comments :

P 19 : replace abundant by artificial

Ln 38 Ref 1 it is not an appropriate reference, in particular to support the present sentence. Indeed the earthquake/tsunami led to environmental damage indeed, but it is not clear what mean the authors. Do they consider it is related to the tsunami or explosion or dispersal of radionuclides. Reformulate to be more less ambiguous.

Ln 38-39 add references to support this comment

Line 60 correct obatined

Figure Need to be enlarged

Line 47-49: I do not understand the sentence (comparison with Cherbonyl) at least “although “ is not appropriate.

Table 1 / Ln 69: add the references of the exposure levels

Ln 72-73 may be better develop , and explain the transition from human to tree. What such focus on forest farmer in the private field. In fact whatever the forest, there is impact, so justify the focus on private forest ?

Regarding the map, it seems samples were not recovered in the main depocenter. Why

Sampling to present in a table (species / sites) for clarity

Ln 118-120 K could be also determined by gamma counting (during the same counting used to determine Cs). Could you provide details of the different methods, how it could be compared , details on 134 and 137 Cs determination , references are also needed.

Ln 139-140 reference needed

LN 148 how the sampling was done , on field ? what about the risk of contamination of samples during sampling ?

Ln 159 why the sampling was different of the one done in April 2012 ?

Table 2 error bars are required. Check the significant digit. If the number in bracktt correspond to th number of samples, it is necessary to provide at least the standard deviation and to indicate it is a mean or median

Figure 2 is great

Table add standard deviation

Ln 248 it is surprising to see radioiodine mentioned whereas there is no iodine result, to suppress

Ln 256-258: when reading this, I realise the authors did not introduce 134/137Cs, I suggest they add few sentences at least the half live of each radionuclides

Section 3.3 it is necessary the authors discuss the measured decrease in ratio compared to the theorical decrease ; the ratios need to ba added in table 4. A figure presenting the measured 137/134Cs and their activities with time and the theorical decay would be great.

4. discussion need to be improved, it is not reasonable to begin this section by several sentences with “could”, first summarize your results and then discuss your hypothesis.

Ln 283 it is really too late to gibe the half-life of each Cs.

Ln 288-294: to improve

Ln 322-323 again no transition for Chernobyl, it is a bad copy and paste from an another draft or does it make sense to compare with Chernobyl, in that xase it is necessary to better explain this.

Ln 295-332 I am not sure it is relevant for a research article. The main interest of this article is the long-term survey of Cs in a forest. A robust discussion of the results need to be done. Then the implication for forest management is more regional and need to be improved or suppressed.

Please complete author contributions and funding

Round 2

Reviewer 4 Report

Data: in fact, mostly presented in tables and as mean values, it is necessary to provide this mean with the standard deviation. It is a shame that the variability of Cs for the same type of samples is not discussed. It must be added as such measurements are not frequent. It could help further works to estimate the appropriate number replicate.

Reference to forest farmers not very clear in the present form. Some parts were really written for this article or recovered from a report/article on Chernobyl. Must be suppressed or improved.

Author Response

Letter to the Fourth Reviewer,

Thank you for reviewing our manuscript.

Data: in fact, mostly presented in tables and as mean values, it is necessary to provide this mean with the standard deviation. It is a shame that the variability of Cs for the same type of samples is not discussed. It must be added as such measurements are not frequent. It could help further works to estimate the appropriate number replicate:

As suggested, we changed Table 4 with this mean with the standard deviation because the trees in Table 4 were cut at the same time (please see attached). 

Table 4. Radiocesium (134Cs and 137Cs) specific activities in trees in the forest (Figure 1:.area number 2) on December 2012.

Tree   species (years)

Ambient   dose

Bark

Xylem

Outer

Inner

Sapwood

Heartwood

Sv/h

kBq   / kg dry weight

Japanese cedar (13)

2.45

8.14 ± 0.32

3.34 ± 0.14

2.13 ± 0.13*

2.60 ± 0.04*

Japanese cedar (13)

2.00

8.90 ± 0.41

5.66 ± 0.19

0.57 ± 0.02*

1.25 ± 0.03*

Japanese cedar (18)

1.88

8.37 ± 0.32

7.29 ± 0.22

0.87 ± 0.04*

1.10 ± 0.03*

Japanese cedar (20)

1.90

14.22 ± 1.00

4.55 ± 0.12

0.58 ± 0.03*

0.99 ± 0.03*

Japanese cedar (29)

2.00

11.36 ± 1.04

4.52 ± 0.11

0.63 ± 0.05

0.86 ± 0.03

Japanese cedar (16)

2.45

12.81 ± 0.92

2.29 ± 0.08

0.36 ± 0.02

0.69 ± 0.03

Red pine (32)

1.64

6.01 ± 0.22

7.87 ± 0.33

1.48 ± 0.23

0.56 ± 0.02

Pine (41)

2.36

14.32 ± 1.11

3.22 ± 0.12

0.65 ± 0.02*

0.51 ± 0.01

Red pine (46)

1.62

14.11 ± 1.22

5.72 ± 0.25

1.09 ± 0.32

0.49 ± 0.01

Cypress (15)

2.54

4.84 ± 0.32

3.23 ± 0.17

0.53 ± 0.02

0.48 ± 0.02

Japanese cedar (30)

2.01

6.05 ± 0.52

4.00 ± 0.18

0.51 ± 0.01

0.42 ± 0.02

Japanese cedar (14)

2.44

11.34 ± 0.94

4.42 ± 0.17

2.66 ± 0.25

0.41 ± 0.01*

Red pine (27)

1.60

7.80 ± 0.33

6.86 ± 0.22

1.01 ± 0.16

0.40 ± 0.01

Japanese cedar (35)

2.00

8.61 ± 0.40

4.87 ± 0.22

0.35 ± 0.02

0.30 ± 0.01

Cypress (14)

2.10

8.65 ± 0.41

5.57 ± 0.27

1.22 ± 0.01

0.30 ± 0.01

Japanese cedar (43)

2.02

9.54 ± 0.45

3.55 ± 0.22

0.28 ± 0.01

0.30 ± 0.01

Cypress (22)

2.02

10.07 ± 0.95

6.54 ± 0.30

1.38 ± 0.07

0.21 ± 0.01

Cypress (16)

2.23

8.58 ± 0.33

7.19 ± 0.32

0.37 ± 0.01

0.05 ± 0.01

All the vascular cambium zones were removed. *P < 0.05 based on Student’s t-test and data are means ± s.d.

Reference to forest farmers not very clear in the present form. Some parts were really written for this article or recovered from a report/article on Chernobyl. Must be suppressed or improved:

I thank you for your comment. However, it is very difficult for me to revise some sentences in the text. Because this comment is very amorphous. When I review some papers, I have always shown the hard sentences definitely. Sometimes, I give my example sentences to the authors.  
